# Randomised, placebo-controlled, phase 3 trial of the effect of the omega-3 polyunsaturated fatty acid eicosapentaenoic acid (EPA) on colorectal cancer recurrence and survival after surgery for resectable liver metastases: EPA for Metastasis Trial 2 (EMT2) study protocol

Mark A Hull [iD] ,[1] Pei Loo Ow,[2] Sharon Ruddock,[2] Tim Brend,[1] Alexandra F Smith,[2] Helen Marshall,[2] Mingyang Song,[3] Andrew T Chan,[4] Wendy S Garrett,[3] Omer Yilmaz,[5] David A Drew,[4] Fiona Collinson,[2] Andrew J Cockbain,[6] Robert Jones,[7] Paul M Loadman [iD] ,[8] Peter S Hall,[9] Catherine Moriarty,[6] David A Cairns,[2] Giles J Toogood[1,6]

For numbered affiliations see end of article.

**Correspondence to**
Professor Mark A Hull;
m.a.hull@leeds.ac.uk

## ABSTRACT

**Introduction** There remains an unmet need for safe and cost-effective adjunctive treatment of advanced colorectal cancer (CRC). The omega-3 polyunsaturated fatty acid eicosapentaenoic acid (EPA) is safe, well-tolerated and has anti-inflammatory as well as antineoplastic properties. A phase 2 randomised trial of preoperative EPA free fatty acid 2 g daily in patients undergoing surgery for CRC liver metastasis showed no difference in the primary endpoint (histological tumour proliferation index) compared with placebo. However, the trial demonstrated possible benefit for the prespecified exploratory endpoint of postoperative disease-free survival. Therefore, we tested the hypothesis that EPA treatment, started before liver resection surgery (and continued postoperatively), improves CRC outcomes in patients with CRC liver metastasis.

**Methods and analysis** The EPA for Metastasis Trial 2 trial is a randomised, double-blind, placebo-controlled, phase 3 trial of 4 g EPA ethyl ester (icosapent ethyl (IPE; Vascepa)) daily in patients undergoing liver resection surgery for CRC liver metastasis with curative intent. Trial treatment continues for a minimum of 2 years and maximum of 4 years, with 6 monthly assessments, including quality of life outcomes, as well as annual clinical record review after the trial intervention. The primary endpoint is CRC progression-free survival. Key secondary endpoints are overall survival, as well as the safety and tolerability of IPE. A minimum 388 participants are estimated to provide 247 CRC progression events during minimum 2-year follow-up, allowing detection of an HR of 0.7 in favour of IPE, with a power of 80% at the 5% (two sided) level of significance, assuming drop-out of 15%.

## STRENGTHS AND LIMITATIONS OF THIS STUDY

⇒ Eicosapentaenoic acid (EPA) for Metastasis Trial 2 is a randomised, double-blind, placebo-controlled trial of 4 g pure EPA ethyl ester daily.

⇒ Treatment starts before colorectal cancer (CRC) liver metastasis surgery and is scheduled for a minimum 2 years.

⇒ The primary endpoint is progression-free survival with a key secondary endpoint of overall survival.

⇒ Participant follow-up and Investigational Medicinal Product delivery can be performed remotely by telephone and courier delivery, which provided COVID-resilience during the pandemic.

⇒ There is a linked biospecimen collection allowing translational studies investigating the mechanism(s) of the anti-CRC activity of EPA.

**Ethics and dissemination** Ethical and health research authority approval was obtained in January 2018. All data will be collected by 2025. Full trial results will be published in 2026. Secondary analyses of health economic data, biomarker studies and other translational work will be published subsequently.

**Trial registration number** NCT03428477.

## INTRODUCTION

Colorectal cancer (CRC) remains the second most common cause of cancer-related death globally with 935 173 deaths registered in 2020, which represents 9.4% of all cancer

deaths worldwide.[1] Only 53% of individuals diagnosed with CRC in England (2013–2017) survive for 10 or more years, with the majority of deaths being related to distant metastasis.[2] Approximately half of all patients with CRC present with synchronous or metachronous liver metastasis (LM).[3] Historically, survival of patients with untreated CRCLM was only a few months.[4] Five-year survival rates for CRCLM patients with contemporary multimodality treatments are now widely of the order of 30%.[5] Surgical resection provides the only prospect of 'cure' for individuals with CRCLM, who are deemed suitable for surgery, with approximately 20% of those undergoing surgery obtaining long-term (>10 years) disease-free survival (DFS).[6] However, overall 5-year survival following liver resection with curative intent and oxaliplatin-based adjuvant chemotherapy is, at best, 60%.[2] Therefore, there remains an unmet need for safe and cost-effective adjunctive treatment for patients with CRCLM.

The omega-3 polyunsaturated fatty acid (PUFA) eicosapentaenoic acid (EPA) is found naturally in highest quantities in oily fish.[7] It is widely used as a nutritional supplement, most often in combination with the other main marine omega-3 PUFA docosahexaenoic acid.[7] Pure EPA ethyl ester is now licensed for use in patients with severe hypertriglyceridaemia and for secondary prophylaxis of vascular events in high-risk individuals taking a statin, who have elevated serum triglyceride levels.[8]

Clinical evidence that EPA has anti-CRC activity at early stages of colorectal carcinogenesis is emerging. Treatment with 99% EPA free fatty acid 2 g daily for 6 months was associated with a significant reduction in rectal adenomatous polyp (the benign precursor of CRC) size (29.8%) and multiplicity (22.4%) compared with placebo in a randomised controlled trial (RCT) in patients with familial adenomatous polyposis undergoing sigmoidoscopic surveillance after total colectomy.[9] The seAFOod polyp prevention trial reported that treatment with EPA free fatty acid 2 g daily was associated with a reduction in adenomatous (but not serrated) colorectal polyp recurrence (as measured by colorectal polyp number) in 'high risk' individuals undergoing colonoscopy surveillance in the English Bowel Cancer Screening Programme.[10]

Omega-3 PUFAs may also modify the natural history of established CRC. Observational data from cohort studies suggest that increased fish intake is associated with improved survival post-CRC diagnosis, the effect being most prominent for proximal (right-sided) CRCs.[11] The EPA for Metastasis Trial (EMT) study was a phase 2 RCT of EPA free fatty acid 2 g daily in patients awaiting liver resection surgery for CRC LM.[12] Although the primary endpoint (the histological CRC tumour cell proliferation index) was null, liver metastases in participants from the active EPA arm had a lower vascularity score than placebo-treated tumours, suggesting possible antiangiogenic activity.[12] In addition, there was an increase in overall survival (OS) and DFS in patients randomised to the EPA arm, both of which were specified as exploratory endpoints based on hypotheses that there is prolonged EPA tissue bioavailability due to the slow tissue 'washout' kinetics of EPA and that oral omega-3 PUFA use alters the human gut microbiome.[13 14]

There is evidence that concurrent omega-3 PUFA therapy may improve the tolerability and efficacy of cancer chemotherapy from a number of small, heterogeneous studies across several gastrointestinal cancers.[15] Two uncontrolled case series support improved tolerability and nutritional parameters in omega-3 PUFA users during FOLFOX and FOLFIRI regimens for CRC.[16 17] Moreover, a study of omega-3 PUFA-containing oral nutritional supplement (ONS) use demonstrated survival benefit in patients with advanced gastrointestinal cancer receiving cancer chemotherapy and a Glasgow score ≥1 compared with individuals who received no ONS.[18] Systematic review and meta-analysis of the effect of omega-3 PUFA-containing ONS intake on cancer cachexia has highlighted the paucity of high-quality evidence but demonstrated a signal towards anti-inflammatory activity and body weight maintenance for omega-3 PUFA supplementation.[19] Therefore, a valid hypothesis is that EPA treatment abrogates the CRC-related cachexia syndrome characterised by fatigue, anorexia and reduced skeletal muscle mass (sarcopaenia).

In summary, given the promising preliminary data supporting anticancer activity of EPA,[20] including possible survival benefit for patients with CRCLM,[12] and the excellent safety and tolerability profile of EPA,[8] we tested the hypothesis that oral treatment with high-dose (4 g daily) pure EPA ethyl ester, started before surgery and continued long-term after liver resection for CRCLM, decreases CRC recurrence and/or improves OS and quality of life, in patients with advanced CRC.

### Trial design

The EMT2 trial is a randomised, double-blind, placebo-controlled, multicentre, parallel-group, superiority phase 3 trial of EPA in patients undergoing liver resection surgery for CRCLM with curative intent. The trial is approved by Newcastle and North Tyneside Research Ethics Committee (16/NE/0140). The trial is based in tertiary-referral Hepatobiliary Surgery Units in England and Wales. Patients are randomised, on an equal basis, to receive either four soft-gel capsules per day each containing 1 g icosapent ethyl (IPE; Vascepa (USA) or Vazkepa (UK and Europe), Amarin Pharma, Bridgewater, New Jersey, USA), taken as two capsules twice daily with food, or four identical placebo capsules (containing pharmaceutical grade mineral oil[21]) taken in an identical fashion, before liver surgery (figure 1). The Investigational Medicinal Product (IMP) is kindly provided by Amarin Pharma and is identical to that used in the REDUCE-IT trial[8] with a daily dose of EPA equivalent to 4 g of EPA-ethyl ester or 3656 mg of EPA free fatty acid). This daily dose of IPE is licensed for use in the USA, UK and Europe for cardiovascular event risk reduction in adult statin-treated patients at high cardiovascular risk with elevated triglycerides (≥150 mg/dL (≥1.7 mmol/L))

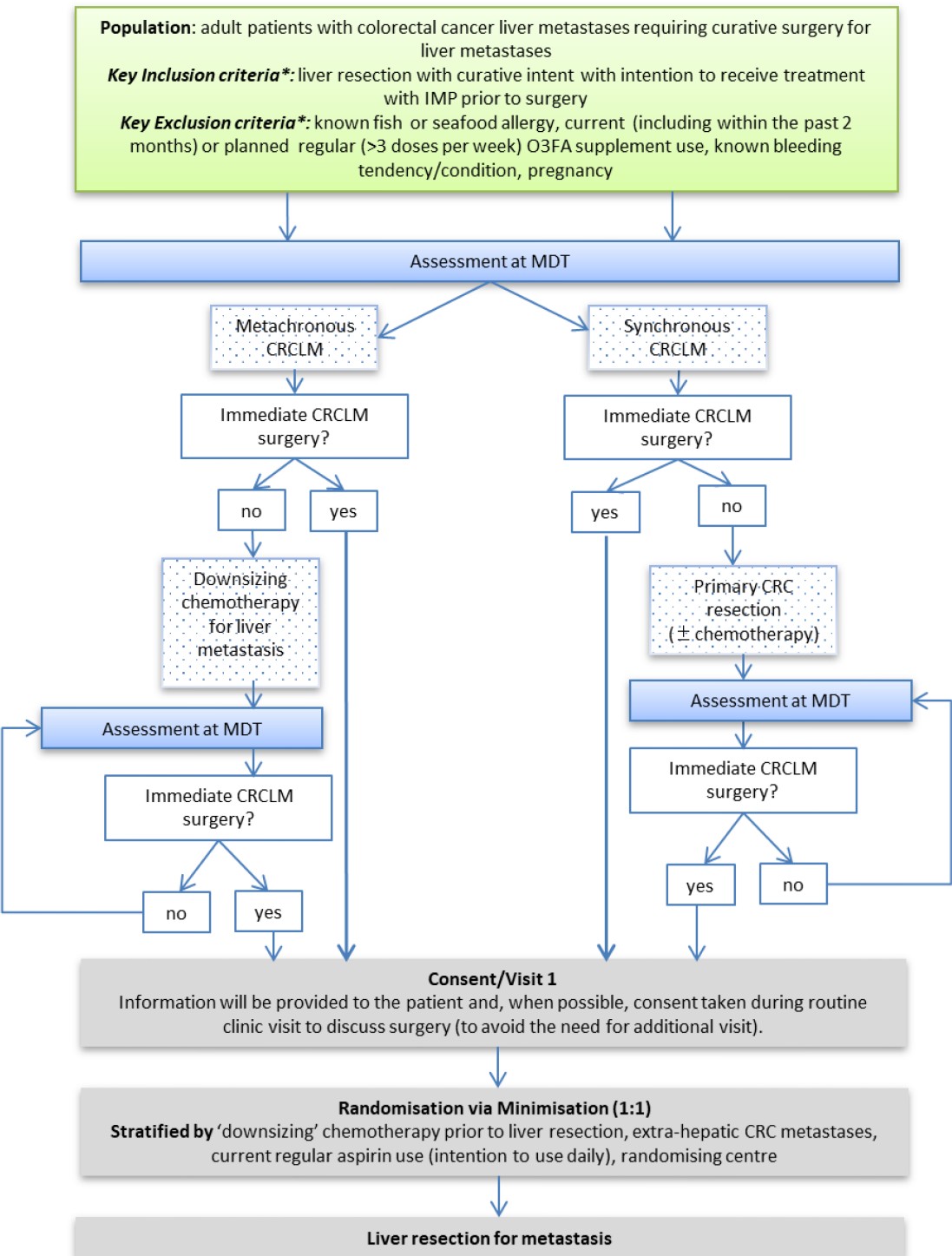

**Figure 1** Pathway for screening, obtaining informed consent and randomisation to the EMT2 trial. *The full list of inclusion and exclusion criteria are listed in table 1. CRC, colorectal cancer; CRCLM, CRC liver metastasis; EMT2, EPA for Metastasis Trial 2; EPA, eicosapentaenoic acid; IMP, Investigational Medicinal Product; MDT, Multi-disciplinary team meeting; O3FA, omega-3 fatty acid.

and either established cardiovascular disease, or diabetes and at least one other cardiovascular risk factor. The use of mineral oil as a clinical trial placebo has been reviewed.[21] In the REDUCE-IT trial, 4 g daily of pharmaceutical grade mineral oil was associated with a small increase in levels of inflammatory and lipid biomarkers associated with atherosclerosis risk,[22] believed to account for, at most,

3% increased cardiovascular risk in the placebo arm of REDUCE-IT.[23] Overall, there is no evidence that mineral oil, at quantities used in clinical trials including EMT2 and REDUCE-IT, impacts significantly on drug absorption or other clinical outcomes in cardiovascular trials.[21]

All participants are scheduled to take IMP for a minimum of 2 years and a maximum of 4 years after liver

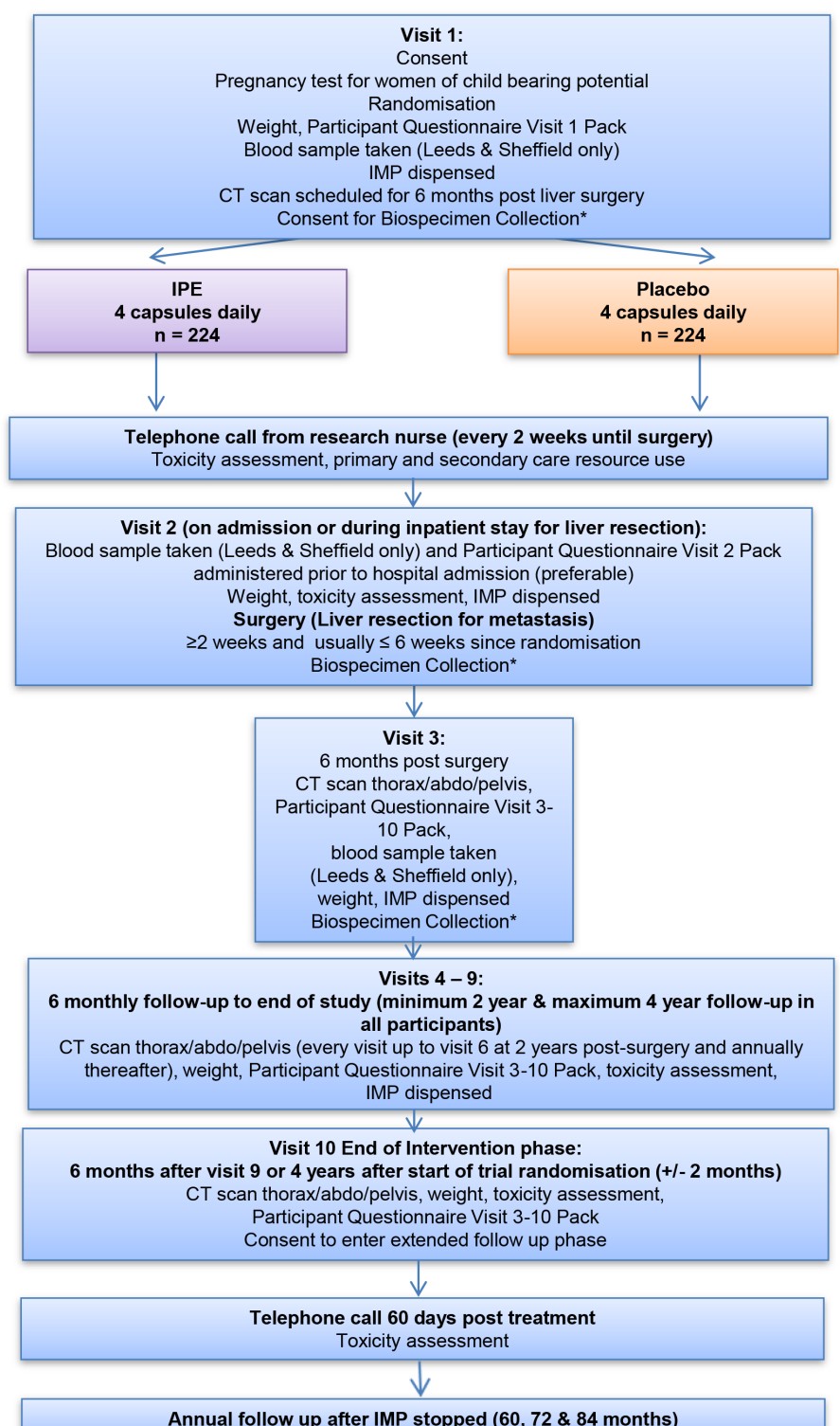

**Figure 2** Assessment schedule from randomisation onwards in the EMT2 trial. *The EMT2 biospecimen collection is a distinct study with separate ethical approval and study registration (see text). EMT2, EPA for Metastasis Trial 2; EPA, eicosapentaenoic acid; IMP, Investigational Medicinal Product; IPE, icosapent ethyl.

resection, with continuing annual clinical record review after the end of the intervention phase (figure 2).

Inclusion and exclusion criteria are listed in table 1. Patients with synchronous primary and liver metastatic CRC are eligible with recruitment possible at the time of the clinical decision to perform CRCLM surgery.

Administration of neoadjuvant or adjuvant chemotherapy is not an exclusion criterion. Exclusion criteria are also consistent with excipients in the IMP capsule. Consistent with the Summary of Product Characteristics (SmPC) for IPE,[24] concurrent use of anticoagulant or antiplatelet medication is allowed.

**Table 1** Inclusion and exclusion criteria for the EMT2 trial

| Inclusion criteria | Exclusion criteria |
|---|---|
| Age ≥18 years | Previous surgery for the management of current CRCLM* |
| Able to provide written informed consent | Incurable extrahepatic metastases |
| Histological diagnosis of CRC with evidence of one or more liver metastases | Current (in the last 2 months) or planned regular (more than three doses per week) use of omega-3 PUFA-containing medication or supplements |
| Planned liver resection surgery for CRCLM with curative intent† | Fish/seafood allergy |
| Intention to receive IMP treatment prior to CRCLM surgery | Hereditary fructose intolerance |
| | Soya or peanut allergy |
| | Inability to comply with trial schedule |
| | Known bleeding tendency (eg, von Willebrand disease) |
| | A previous malignancy within 5 years‡ |
| | Pregnant/breastfeeding woman or WOCBP not willing to use effective contraceptive measures§ |
| | Fertile (postpubescent and not permanently sterile by vasectomy or bilateral orchidectomy) man not willing to use barrier contraception |

*Patients who have already undergone one CRCLM surgery as part of a two-stage procedure are ineligible.
†Includes repeat CRCLM liver surgery (a second independent liver resection) for a separate metachronous CRC liver recurrence.
‡Other than CRC, non-melanoma skin cancer where treatment consisted of resection only or radiotherapy, ductal breast carcinoma in situ where treatment consisted of resection only, cervical carcinoma in situ where treatment consisted of resection only, and superficial bladder carcinoma where treatment consisted of resection only.
§WOCBP: Effective contraception includes combined (oestrogen containing and progesterone containing) hormonal contraception, progesterone-only hormonal contraception, intrauterine device, intrauterine hormone-releasing system, bilateral tubal occlusion, vasectomised partner, practising true sexual abstinence.
CRC, colorectal cancer; EMT2, EPA for Metastasis Trial 2; EPA, eicosapentaenoic acid; IMP, Investigational Medicinal Product; PUFA, polyunsaturated fatty acid; WOCBP, woman of childbearing potential.

The eligibility of individuals, who are participating in another trial, is determined on a case-by-case basis by the research site team, clinical trials unit (CTU) and sponsor. Co-enrolment with participants in the CRC cohort of the Add-Aspirin trial (ISRCTN74358648) is permitted providing that the respective eligibility criteria of each trial are fulfilled.[25]

The current protocol is version 9.0 dated 27 July 2022. The trial website is https://ctru.leeds.ac.uk/EMT2/.

## Trial endpoints
### Primary endpoint
The primary endpoint is progression-free survival (PFS) during a minimum of 2 years follow-up. PFS is defined as (1) the time from randomisation to death (from any cause), (2) the first documented evidence of CRC progression (defined as the date of the CT scan or the relevant assessment, at which disease progression or new recurrence is identified and which can be clinical progression or radiological progression evaluated by RECIST V.1.1[26]) or (3) new CRC recurrence or clinical deterioration unequivocally due to CRC progression. Participants without a CRC progression event will be censored at the time of the last assessment when they were alive and progression-free.

### Secondary and exploratory endpoints
The key secondary endpoint is OS, which is defined as the time from randomisation to death from any cause. Participants alive at analysis will be censored at the time of the last assessment when they were alive.

Other secondary endpoints are (1) the safety and tolerability of IPE 4 g daily before and after surgery, including during cancer chemotherapy; (2) patient-reported quality of life measures; (3) cost-effectiveness of IPE assessed using a modified UK Cancer Costs Questionnaire Version 2.0 and (4) new primary cancer (excluding non-melanoma skin cancer, breast ductal cancer in situ, cervical carcinoma in situ and superficial bladder carcinoma).

An exploratory endpoint is the red blood cell (RBC) membrane EPA level measured at baseline (visit 1), presurgery (visit 2) and 6 months after CRCLM surgery (visit 3) from blood samples taken from two Yorkshire research sites (Leeds and Sheffield).[10] In addition, the skeletal muscle area at the L3 level on CT imaging at 6 months after CRCLM may be measured as another exploratory endpoint dependent on the availability of additional, specific funding for that analysis.

### Trial schedule
Potential participants are identified by the hepatobiliary multidisciplinary team (MDT) at the time of the decision to offer CRCLM surgery with curative intent (figure 1). Potential participants who require a primary CRC resection or 'downsizing' neoadjuvant chemotherapy for CRCLM prior to surgery are reviewed by the MDT again, following primary management, in order to confirm up-to-date eligibility. A woman of childbearing potential is required to undergo a pregnancy test prior to randomisation.

After written informed consent is obtained, participants are allocated 1:1, using minimisation with a random element, to active or placebo treatment up to the day before CRCLM surgery using an automated 24-hour telephone/online randomisation service based in Leeds CTU. Minimisation is for (1) neoadjuvant chemotherapy prior to liver resection; (2) known extrahepatic CRC metastases with curative treatment planned; (3) current regular aspirin use or Add-aspirin trial participant still taking IMP; (4) research site and (5) time between randomisation and date of planned liver resection surgery either <2 weeks or ≥2 weeks.

The schedule of trial assessments before CRCLM surgery, in the perioperative period and postoperatively is described in figure 2 and online supplemental figure 1. Assessments are carried out every 2 weeks between randomisation and CRCLM surgery by telephone and on the first day of the hospital stay for CRCLM surgery. Postoperative assessments are 6 monthly with a minimum CT Thorax/Abdomen/Pelvis every 6 months for 2 years followed by annual CT imaging thereafter, in line with routine clinical practice at most sites. Quality of life (QoL) is assessed using EQ-5D-5L, EORTC QLQ-C30 and QLQ-LMC21 supplementary module questionnaires. Health economic outcomes are assessed using a modified UK Cancer Costs Questionnaire Version 2.0 at each trial visit and telephone call.

Participants are followed up for 60 days beyond the end of IMP treatment, followed by annual case record review in order to collect CRC progression and OS data.

### Sample size

Prior to the trial opening, 448 participants (224 per arm) were anticipated to give 247 CRC progression events that are required to detect an HR of 0.7 in favour of the treatment arm with a power of 80% at the 5% (two-sided) level of significance. The sample size estimate assumed that the control arm would have a median PFS of 21 months and the treatment arm would have a median PFS of 30 months,[27 28] with recruitment in a 2-year period, minimum 2-year follow-up per participant and with 10% drop-out.

The HR for PFS in the phase II EMT study was 0.694,[12] but the EPA arm, by chance, recruited a higher proportion of patients with poor prognostic features, and a multivariate analysis including these factors actually yielded an HR of 0.35 (95% CI 0.15 to 0.79) lending credence to the assumed more conservative HR of 0.7 chosen for this larger phase three study, given that the magnitude of benefit observed in phase two studies is often greater than that observed in the phase III counterpart.

The EMT study tested EPA-FFA 2 g daily.[12] Although the EMT2 study is evaluating IPE 4 g daily, we have assumed at least similar pharmacokinetics of EPA given as IPE, compared with lower-dose EPA-FFA.

Slower than expected recruitment and a higher withdrawal rate due to the COVID-19 pandemic and other factors (see the Trial progress section) prompted re-evaluation of the sample size required to generate 247 CRC progression events during an extended recruitment period (5.5 years), which provides longer cumulative follow-up. We used a generalised F distribution, rather than an exponential function, which more accurately reflected blinded, live PFS data. A simulation study (1000 simulations with random selection of 90% of participants to account for drop-out) in October 2021 predicted that a minimum 388 participants would be required to observe 247 CRC progression events (accounting for actual participant withdrawals, including those related to a pause of the trial at one research site during the first COVID-19 lockdown).

### Treatment

Participants receive either four 1 g capsules of IPE or four identical placebo capsules daily (two capsules two times per day with food) from the date of randomisation until the participant has completed study treatment. Treatment is scheduled to continue after disease progression or diagnosis of a new primary cancer. Treatment compliance is assessed at each trial visit or telephone call to determine if the participant has delayed, missed or modified dosing. Unused, expired capsules are collected from participants at a trial visit or at the end of the scheduled intervention period, for counting.

A participant is allowed to take one or more treatment breaks during the trial follow-up, if the participant experiences a persistent adverse reaction (AR), or difficulty taking IMP (eg, because of restrictions to oral intake, during CRC chemotherapy or due to other concurrent treatment). In the event of a persistent AR, dose reduction (to two capsules daily) is recommended prior to a complete trial treatment break.

'Over-the-counter' nutritional supplements containing any omega-3 PUFA or prescribed omega-3 PUFAs (eg, Omacor) are prohibited while participants are receiving trial treatment and are an exclusion criterion for the EMT2 trial. Any participant consistently using another omega-3 PUFA preparation is withdrawn from trial treatment. Concomitant use of aspirin, other anti-platelet agents or anticoagulants such as warfarin or a direct-acting oral anticoagulant are allowed, in keeping with the SmPC for IPE (Vazkepa).[24]

### Safety

Reference safety information with which to assess seriousness, expectedness and causality for adverse events was an Investigator Brochure for IPE (Vascepa) version 2 dated 1 June 2017 until January 2022, after which the SmPC for IPE (Vazkepa)[24] was used after the MHRA granted a licence for use of IPE for secondary prophylaxis of vascular events in high-risk individuals taking a statin, who have elevated serum triglyceride levels.

### Statistical analysis

A full statistical analysis plan will be written before any analysis is undertaken. Analysis will be conducted on an intention-to-treat basis, in which participants will be included according to the treatment they were randomised to receive, apart from safety/tolerability endpoints which will be based on a safety population of all patients who took at least one dose of any trial treatment.

All hypothesis tests will be two sided and will use a 5% significance level. No interim futility or early stopping analyses are planned. The final analysis will only take place after the last patient recruited has been followed up for a minimum 24 months, has withdrawn or died.

The primary endpoint (PFS) and the key secondary endpoint (OS) will be reported using the 95% CI of the HR. PFS and OS curves will be calculated using the Kaplan-Meier method. Differences in PFS and OS between the

treatment groups will be compared using multivariate modelling to adjust for the minimisation factors. Treatment HRs and corresponding 95% CIs will be obtained from the multivariate models.

A subgroup analysis comparing the outcomes for patients that receive ≥2 weeks of IMP prior to surgery and patients that receive <2 weeks of EPA will be performed. Sensitivity analyses for PFS and OS will be conducted to examine the effects of other potential modifiers, in addition to the minimisation factors. Other sensitivity analyses may be carried out for each endpoint to take into account differing assumptions about missing data, if there is a significant amount of missing data. Further subgroup and sensitivity analyses may also be performed and will be detailed in the full statistical analysis plan.

Safety information will be summarised by treatment received. The suspected relationship to IPE will be presented along with other causality, the outcome and the event duration. The number and timing of new primary cancers will also be summarised descriptively by treatment group.

QoL measures will be analysed using random effects (multilevel) models to account for the hierarchical nature of repeated measures data and the models will include adjustments for baseline QoL and the minimisation factors. QoL will be presented using mean scores along with the 95% CI. Similar summaries will be produced for quality-adjusted life-years (QALYs), as scored by the EQ-5D-5L questionnaire. Cost-effectiveness comparisons between treatment arms will be analysed using the incremental cost-effectiveness ratio. The cost per disease recurrence prevented at 2 years and the cost per QALY at 2 years will be reported.

The relationship between the RBC membrane EPA level and both PFS and OS will be analysed using an extended Cox model, which will adjust for the minimisation factors and will also include the RBC membrane EPA level (% of total fatty acids)[10] as a time-dependent covariate.

### Trial progress

Health research authority approval (Newcastle and North Tyneside REC 16/NE/0140 and MHRA CTA 16767/0289/) was granted on 3 April 2017. EMT2 was prospectively registered with ClinicalTrials.gov (NCT03428477). The European Clinical Trials Database registration is EudraCT 2016-000628-24.

The EMT2 trial opened to recruitment in March 2018. First participant's first visit was 2 May 2018. The original recruitment period was estimated to be 24 months based on a recruitment projection of 18 patients per month by 8 research sites. This recruitment estimate was based on the annual CRCLM surgery volume at the research sites in 2014 (890 procedures) and a 25% randomisation rate of screened patients. However, the mean monthly recruitment in the first 12 months was four per month. In the early phase of recruitment, 38% of potential participants undergoing screening were ineligible because CRCLM surgery was scheduled within 2 weeks of randomisation.

This prompted a substantial protocol amendment to allow recruitment of patients without stipulation about the length of IMP treatment before surgery (online supplemental data).

The number of research sites has since been expanded to 13 (see online supplemental data for the list of research sites and their date of opening for recruitment).

The EMT2 trial did not recruit any participants for approximately 2 months at the onset of the COVID-19 pandemic restrictions in the UK in early 2020, although, in contrast with many randomised trials in the UK, the trial remained open, except for one NHS Trust research site that paused all non-COVID-19 research activity for several months from March 2020. During the COVID-19 pandemic, recruitment continued with wide variations in monthly recruitment rates related to research staff availability (affected by secondment to COVID-19 research roles and sickness), alterations to routine hepatobiliary surgery activity and practice, and patient acceptability. The ability to deliver IMP by post and perform trial follow-up visits by telephone aided trial acceptability and retention during the pandemic. We did not observe an increase in trial withdrawal during national COVID-19 lockdowns between March 2020 and mid-2022.

Based on a revised recruitment projection of 5 participants per month, in order to reach a revised recruitment target of 388, the recruitment period has been extended to 67 months (end of recruitment November 2023) with a minimum 2-year follow-up. In order to collect the maximum number of CRC progression events from randomised individuals, the protocol was amended to obtain consent from participants at the end of their 4-year intervention period to collect data on any CRC progression event from their healthcare record on an annual basis until the end of the follow-up period of the EMT2 trial at the end of November 2025.

Other protocol amendments during EMT2 trial conduct are listed in online supplemental data. Key changes during the trial were removal of the stipulation that IMP was started more than 2 weeks before CRCLM surgery (prompted by realisation that the majority of surgeries were performed within 2 weeks of the decision to offer CRCLM resection; SA06), and addition of new exclusion criteria related to peanut/soya allergy and sorbitol/xylitol intolerance (SA14).

### EMT2 biospecimen collection and ancillary translational studies

Separate funding was obtained to collect faeces, urine, blood and tumour tissue from EMT2 participants after randomisation to the EMT2 trial for translational laboratory studies. This work is performed under separate HRA approval (REC 20/YH/0306) and ClinicalTrials.gov registration (NCT04682665) as the EMT2 biospecimen collection. The current protocol for Biospecimen collection is version 3.0 dated 28 April 2022.

The EMT2 biospecimen collection is part of a collaboration between the University of Leeds and researchers

across several institutions in Boston, USA (Massachusetts General Hospital (MGH), Harvard TH Chan School of Public Health and Massachusetts Institute of Technology) and the University of Bradford, funded by National Institutes of Health to investigate the role of the gut microbiota and its relationship with the host antitumour immune response in the anti-CRC activity of EPA.

Consent to provide biospecimens is obtained from new EMT2 participants, only after EMT2 entry. Faecal, blood and urine samples are collected (1) after EMT2 trial randomisation, before starting active EPA or placebo, (2) just before surgery and (3) at 6 months after CRCLM surgery (online supplemental figure 2), with the option to provide further specimens during EMT2 follow-up out to 4 years. It is also possible for existing EMT2 participants to provide a 'one-off' faecal and urine sample at any time while taking IMP during the follow-up period. There are five participating sites, from which baseline (visit 1) and surgery (visit 2) samples are transferred to the Biobank in Leeds. We use a modified Micro-N collection tool,[29] which facilitates collection of faecal specimens in a DNA Genotek OMNIgene.GUT tube (for metagenome and metatranscriptome profiling), in 95% ethanol (for faecal metabolomic analysis), and an anaerobic tube filled anaerobically with liquid dental transport medium (for culture and gnotobiotic animal model studies), for postal delivery to the Leeds Biobank, alongside a urine sample (online supplemental figure 2).

Laboratory studies will take place in Boston, Leeds and Bradford (online supplemental figure 3). Linked, fully anonymised data (including EMT2 survival outcomes) will be shared between Leeds and MGH using the EMT2 trial Safety Statistician as the 'honest broker' in order to avoid inadvertent disclosure of individual treatment allocation to EMT2 Investigators (online supplemental figure 3).

## Patient and public involvement

A patient representative was involved in trial design and is a member of the trial management group. This patient representative has supported design of all patient-facing materials including website content and Instructions for biospecimen collection. A different patient representative sits on the trial steering committee.

### Author affiliations
[1]Leeds Institute of Medical Research, University of Leeds, Leeds, UK
[2]Leeds Cancer Research UK Clinical Trials Unit, Leeds Institute of Clinical Trials Research, University of Leeds, Leeds, UK
[3]Harvard TH Chan School of Public Health, Boston, Massachusetts, USA
[4]Clinical and Translational Epidemiology Unit, Massachusetts General Hospital, Boston, Massachusetts, USA
[5]Koch Institute for Integrative Cancer Research at Massachusetts Institute of Technology, Boston, Massachusetts, USA
[6]Leeds Teaching Hospitals NHS Trust, Leeds, UK
[7]Royal Liverpool and Broadgreen University Hospitals NHS Trust, Liverpool, UK
[8]Institute of Cancer Therapeutics, University of Bradford, Bradford, UK
[9]Edinburgh Clinical Trials Unit, University of Edinburgh, Edinburgh, UK

**Acknowledgements** The Trial Management Group wish to thank Amarin Pharma for providing active and placebo IMP for the EMT2 trial. Amarin Pharma played no role in the design or conduct of the EMT2 trial. The support of the Clinical Trials Research Unit (CTRU) at the University of Leeds is essential to the successful running of the study; we thank all CTRU staff who have contributed, past and present. The Trial Management Group also wish to acknowledge the important role played by the Patient and Public Representatives on the Trial Management Group and Trial Steering Committee.

**Contributions** MAH is the chief investigator. MAH: conceptualisation, investigation, resources, writing–original draft, writing–review and editing, supervision, project administration, funding acquisition PLO: methodology, formal analysis, investigation, data curation, writing–original draft, writing–review and editing SR: investigation, writing–review and editing, visualisation TB: investigation, resources, data curation, writing–review and editing. AFS: methodology, investigation, data curation, writing–review and editing, supervision, project administration HM: methodology, formal analysis. MS: conceptualisation, resources, writing–review and editing, funding acquisition. ATC: conceptualisation, resources, writing–review and editing, supervision, project administration, funding acquisition. WSG: conceptualisation, investigation, resources, writing–review and editing, funding acquisition OY: conceptualisation, investigation, resources, writing–review and editing, funding acquisition DAD: investigation, data curation. FC: conceptualisation, writing–review and editing, funding acquisition. AJC: investigation, resources, writing–review and editing. RJ: investigation, resources, writing–review and Editing PML: methodology, investigation, writing–review and editing, supervision, funding acquisition. PSH: methodology, resources, writing–review and editing, funding acquisition. CM: investigation, writing–review and editing. DAC: methodology, formal analysis, investigation, data curation, writing–review and editing, supervision, project administration. GJT: conceptualisation, investigation, resources, writing–review and editing, supervision, project administration, funding acquisition.

**Funding** The funder and the Sponsor had no role in the design of the trial, data collection, or input into the writing of this manuscript.The EMT2 trial is funded by Yorkshire Cancer Research (L387) and is sponsored by the University of Leeds. The EMT2 biospecimen collection is funded by the National Institutes of Health (1R01CA243454-01A1) and is sponsored by the University of Leeds (governance-ethics@leeds.ac.uk). Both studies have been adopted to the NIHR Clinical Research Network (CRN) Portfolio (CPMS ID 34700 and 47372, respectively) and have benefited from CRN research staff support.

**Competing interests** None declared.

**Patient and public involvement** Patients and/or the public were involved in the design, or conduct, or reporting, or dissemination plans of this research. Refer to the Methods section for further details.

**Patient consent for publication** Not applicable.

**Provenance and peer review** Not commissioned; externally peer reviewed.

#### ORCID iDs
Mark A Hull http://orcid.org/0000-0001-7414-1576
Paul M Loadman http://orcid.org/0000-0002-4259-8616

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
