## [Reviewer comments · BMJ Open]

ARTICLE DETAILS

TITLE (PROVISIONAL)	A randomised, placebo-controlled, phase 3 trial of the effect of the omega-3 polyunsaturated fatty acid eicosapentaenoic acid (EPA) on colorectal cancer recurrence and survival after surgery for resectable liver metastases: EPA for Metastasis Trial 2 (EMT2) study protocol
AUTHORS	Hull, Mark; Ow, Pei Loo; Ruddock, Sharon; Brend, Tim; Smith, Alexandra; Marshall, Helen; Song, Mingyang; Chan, AT; Garrett, Wendy; Yilmaz, Omer; Drew, David; Collinson, Fiona; Cockbain, Andrew; Jones, Robert; Loadman, Paul; Hall, Peter; Moriarty, Catherine; Cairns, David; Toogood, Giles

VERSION 1 – REVIEW

REVIEWER	Calder, Philip University of Southampton, Institute of Human Nutrition
REVIEW RETURNED	18-Aug-2023

GENERAL COMMENTS	This manuscript describes the protocol for an RCT of high dose eicosapentenoic acid (3.6 g/day) looking primarily at colorectal cancer recurrence and survival after surgery for resectable liver metastases. There are a number of secondary and exploratory outcomes. The study is funded, approved and underway. Recruitment has been slow and there have been some COVID-related challenges. These are all documented. The rationale for doing the study is robust and is well documented in the Introduction. There is debate in the cardiovascular literature about the placebo being used (mineral oil) and it would be helpful for the authors to add a small section justifying the use of the placebo and mentioning the debate.
--

REVIEWER	Jia, Yongliang Zhengzhou University
REVIEW RETURNED	07-Oct-2023

GENERAL COMMENTS	This manuscript is a significant study and can be considered for publication.
---

VERSION 1 – AUTHOR RESPONSE

Reviewer 1

Prof. Philip Calder, University of Southampton

This manuscript describes the protocol for an RCT of high dose eicosapentenoic acid (3.6 g/day)

looking primarily at colorectal cancer recurrence and survival after surgery for resectable liver metastases. There are a number of secondary and exploratory outcomes. The study is funded, approved and underway. Recruitment has been slow and there have been some COVID-related challenges. These are all documented. The rationale for doing the study is robust and is well documented in the Introduction. There is debate in the cardiovascular literature about the placebo being used (mineral oil) and it would be helpful for the authors to add a small section justifying the use of the placebo and mentioning the debate.

The use of pharmaceutical grade mineral oil in the placebo IMP was already described with a reference to a 2020 review of the use of mineral oil in clinical trials (reference #21). Additional sentences have been added to the text (*Trial Design* Page 7 line 16 onwards) describing the main findings relevant to EMT2 IMP use and containing a conclusion that there is no evidence that mineral oil, at doses used in clinical trials, affects clinical outcomes. An extra two references have been added.

Reviewer 2

Dr. Yongliang Jia, Zhengzhou University, The Second Affiliated Hospital of Zhengzhou University

This manuscript is a significant study and can be considered for publication.